# Synthetic Programming Elicitation for Text-to-Code in Very Low-Resource Programming and Formal Languages

Federico Mora[1]    Justin Wong[1]    Haley Lepe[2]    Sahil Bhatia[1]    Karim Elmaaroufi[1]
George Varghese[3]    Joseph E. González[1]    Elizabeth Polgreen[4]    Sanjit A. Seshia[1]

[1]UC Berkeley    [2]Stanford University    [3]UCLA    [4]University of Edinburgh

{fmora, justin.wong, sahilbhatia, k.e, jegonzal, sseshia}@berkeley.edu
halepe@stanford.edu, varghese@cs.ucla.edu, elizabeth.polgreen@ed.ac.uk

## Abstract

Recent advances in large language models (LLMs) for code applications have demonstrated remarkable zero-shot fluency and instruction following on challenging code related tasks ranging from test case generation to self-repair. Unsurprisingly, however, models struggle to compose syntactically valid programs in programming languages unrepresented in pre-training, referred to as very low-resource Programming Languages (VLPLs). VLPLs appear in crucial settings, including domain-specific languages for internal tools, tool-chains for legacy languages, and formal verification frameworks. Inspired by a technique called natural programming elicitation, we propose designing an intermediate language that LLMs "naturally" know how to use and which can be automatically compiled to a target VLPL. When LLMs generate code that lies outside of this intermediate language, we use compiler techniques to repair the code into programs in the intermediate language. Overall, we introduce *synthetic programming elicitation and compilation* (SPEAC), an approach that enables LLMs to generate syntactically valid code even for VLPLs. We empirically evaluate the performance of SPEAC in a case study for the UCLID5 formal verification language and find that, compared to existing retrieval and fine-tuning baselines, SPEAC produces syntactically correct programs more frequently and without sacrificing semantic correctness.

## 1  Introduction

Large language models (LLMs) have demonstrated an exceptional ability to generate code from natural language prompts for popular programming languages, like Python and Java [13]. Unfortunately, these same language models struggle to generate code for low-resource programming languages, like many domain-specific languages (e.g., CUDA [40]). These challenges are even more pronounced for very low-resource programming languages (VLPLs) (e.g., formal verification languages like UCLID5 [34, 37]). Existing work has attempted to remedy this issue through prompting, constrained decoding, and fine-tuning strategies. Unfortunately, these approaches fail to capture the intricacies of real VLPLs and so success remains limited. To see why, consider the following three exemplars.

First, Wang at el. [42] include context-free grammars in text-to-code prompts to guide LLMs toward syntactically correct answers. This approach works well for simple languages but cannot capture many programming languages that are context-sensitive. Second, Agrawal et al. [1] use static analysis techniques to reject tokens that lead to syntactically incorrect output programs. This technique can go beyond context-free languages but assumes a linear programming process. Unfortunately, it is well known that the root cause of a programming error need not surface as it is written [32], necessitating

38th Conference on Neural Information Processing Systems (NeurIPS 2024).

backtracking and a nonlinear programming process. Third, Cassano et al. [10] translate training data from high resource languages to low resource languages and then use this new data to fine-tune models. This approach is restricted to languages where the LLM is able to translate to the language reliably but unable to generate code from natural language. Further, this approach makes the overly restrictive assumption that the target low-resource language is general purpose: e.g., we cannot translate arbitrary Java programs to CUDA.

In this paper, we propose a text-to-code approach that is fundamentally different (and complementary) to prompting, decoding, and fine-tuning strategies. The first key idea behind our approach comes from natural programming elicitation, a kind of study that helps programming language designers understand how programmers "naturally" approach problems from a given programming domain [29, 11]. Programming language designers use the results of these studies to create languages that are aligned with the expectations of users, leading to less programming friction and more effective developers. We borrow this idea for the setting where LLMs are the "users" of programming languages. Akin to uncovering what human users find "natural" for a given domain, we uncover what LLMs find "natural." Specifically, our first insight is to embrace LLM's tendencies and design an intermediate language that aligns with these LLM expectations.

The second key idea in our approach is that program analyses and repair that are overly aggressive for human users may be suitable for LLM "users." For example, in UCLID5, all variables have to be declared and statically typed: an assignment like `x = 0;` would require a corresponding declaration like `var x:   integer;`. But, if an LLM generates code that had an assignment without a declaration, instead of crashing, one could automatically "repair" the program and output the result.

We use these two ideas to define a new text-to-code approach called *synthetic programming elicitation and compilation* (SPEAC, pronounced "speak"). Specifically, for a target VLPL $T$, we use synthetic programming elicitation to select an intermediate language $P$ (the "parent" language) and define a subset of the language $C$ (the "child" language). The language $P$ should be one that LLMs are good at generating (e.g. Python); the language $C$ should be easy to compile to the target VLPL $T$. Our approach takes $P$, $C$, and a compiler from $C$ to $T$, and produces a text-to-code pipeline for the VLPL $T$. This pipeline uses deductive techniques to automatically repair programs generated by LLMs that are in $P$ but not in $C$. When these deductive techniques are unable to fully repair a program, we insert a "hole" and ask an LLM to finish the repair, repeating as necessary.

We demonstrate the effectiveness of this idea by implementing a prototype, called Eudoxus, that targets UCLID5 [34, 37], a language used for formal modeling and verification of state transition systems. UCLID5 has code examples numbering in the hundreds rather than thousands or millions. Furthermore, UCLID5 programs rely heavily on the notion of a transition system, which is not frequently found in other programming languages. As such, state-of-the-art LLMs are unable to generate any useful UCLID5 code out-of-the-box (see §5.1). In our case study, we use Python as the parent language $P$ and a subset of Python as the child language $C$, and improve the performance of LLM code generation for UCLID5.

Overall, we make the following contributions: 1) We present SPEAC, a novel method for generating syntactically correct code from LLMs in very low resource programming languages; 2) We implement this method for the UCLID5 verification language; and 3) We demonstrate substantial improvement with SPEAC in syntactic correctness, producing parsable code in UCLID5 84.8% of the time compared to 9.1% by gpt-3.5-turbo fine-tuning and 12.1% by gpt-4-turbo in-context learning.

## 2   Related Work

**LLMs for Code Generation.** Modern language models perform exceptionally well on natural language to code generation tasks. For example, proprietary models like GPT-4 [31], the Gemini [41] and Claude 3 Families [1], and open-source models such as Code-Llama [35] and Deepseek-Coder [20] have achieved impressive success on benchmarks such as HumanEval [13], Mostly Basic Python Problems (MBPP) [3], and LiveCodeBench [22]. However, LLMs perform better on popular programming languages that are well represented in training sets (e.g., Python) than less popular programming languages (e.g., Bash) [20, 26, 33]. This effect is even more pronounced for very low resource programming languages, like UCLID5, as shown in §6.

---

[1]https://www.anthropic.com/news/claude-3-family

| Model a state machine describing a traffic light at a pedestrian crosswalk. This is a time triggered machine that assumes it will react once per second. It starts in the red state and counts 60 seconds with the help of the variable count. It then transitions to green, where it will remain until the pure input pedestrian is present. That input is generated by some other subsystem that detects when a pedestrian is present, and should be modelled as nondeterministic. When pedestrian is present, the machine transitions to yellow if it has been green for at least 60 seconds. Otherwise, it transitions to pending, where it stays for the remainder of the 60 second interval. This ensures that once the light goes green, it stays green for at least 60 seconds. At the end of 60 seconds, it will transition to yellow ... | |
|---|---|

```
module TrafficLight {                                      1
  var sigG, sigR, sigY: boolean;                           2
  var count, state: integer;                               3
  var pedestrian: boolean;                                 4
  init {                                                   5
    sigG = false; sigY = false; sigR = true;               6
    state = 0; count = 0; pedestrian = false; }            7
  procedure step()                                         8
    modifies sigG; modifies sigY; modifies sigR;           9
    modifies count; modifies state; {                      10
    if (state == 0) {                                       11
      sigG = false; sigY = false; sigR = true;             12
      count = 0;                                            13
      if (count < 60) { count = (count + 1); }             14
      else { state = 1;}}                                   15
    else { if (state == 1) {                                16
      sigR = false; sigY = false; sigG = true;             17
      if (pedestrian) {                                     18
        if (count >= 60) { state = 2; }                     19
        else { state = 3; }}}}                              20
...                                                         21
```

(a)                                                             (b)

Figure 1: Partial task description from Lee and Seshia [23] (a) and partial output of Eudoxus in UCLID5 (b). We interpret `sigG`, `sigR`, `sigY` to represent green, red, and yellow light signals, respectively. The procedure `step` captures the transition relation of the state machine. `state` appears to be a bookkeeping variable that is used to track the cases in the task, and `count` represents a timer.

**Training Free Approaches for Low-Resource Programming Languages.** In constrained decoding, syntactically incorrect code is avoided by rejecting impossible prefixes, without producing the full code [18, 1, 36, 27]. In the context of autoregressive LLMs that naturally produce code left to right, it remains an open problem how to best include the inductive bias of a grammar. Bhatia et al. [8] use LLMs to rewrite existing code into domain-specific languages and prove that the translation is correct. Misu et al. [28] use retrieval-augmented generation to write Dafny code, another low-resource language for verification [25]. Unlike these works, we focus on very low resourced languages (VLPLs), which have far fewer training examples in the public domain. Our approach is most similar to techniques that allow LLMs to hallucinate but iteratively repair errors [38, 14, 30]. For example, Elmaaroufi et al. [16] use a mixture of prompting and compiler feedback to generate and iteratively repair Scenic code—a probabilistic VLPL that looks like Python [17].

**Training LLMs for Low-Resource Programming Languages.** Recent work has considered augmenting LLMs with support for low-resource programming languages [10, 12, 20]. Chen et al. [12] show that, on smaller 125M parameter encoder-only models, fine-tuning on adjacent languages improves the monolingual performance coding tasks. Synthetic fine-tuning datasets curated and cleaned by LLMs have shown promise for programming tasks. For example, Cassano et al.[10] targets low-resource programming languages (e.g., Julia), using an LLM to translate code examples from high-resource languages to the low-resource language. This process is promising for cases where the language model already has a baseline knowledge necessary to translate to the low-resource language and the target language is general purpose, which is not always the case for VLPLs.

## 3   Overview and Running Example

Given a natural language description of a programming task and a target programming language $T$, the text-to-code problem is to generate a program $t \in T$ that satisfies the task specification. Fig. 1 shows a real input-output pair generated by an instance of our approach targeting the UCLID5 programming language. Specifically, Fig. 1a shows a task extracted from Lee and Seshia [23], and Fig. 1b shows the output corresponding to that task using a prototype implementation of our approach. Fig. 1b passes all compiler checks but has a subtle semantic mistake on line 4.

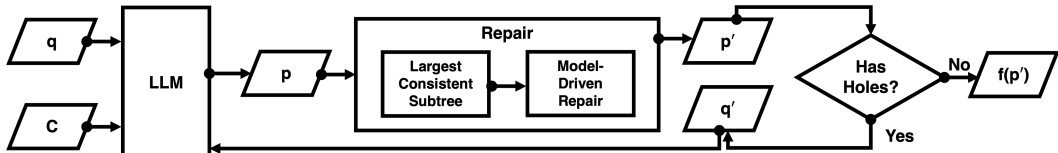

Figure 2: The SPEAC workflow. Users input $q$, a task in natural language, and $C$, a description of the intermediate language. The LLM takes these inputs and generates $p$, a program in $P$. We use formal techniques to repair $p$ and produce $p'$, a program in $C$ that possibly contains holes. If $p'$ does not contain holes, SPEAC applies $f$, a compiler from $C$ to the target language, $T$, and returns the result. Otherwise, SPEAC generates a new prompt, $q'$, and repeats by asking the LLM to fill in the holes.

Fig. 2 shows the workflow that generates models based on task descriptions as shown in Fig. 1b. The workflow is parameterized by an LLM, $L$ (e.g., gpt-3.5-turbo-0125); a target language, $T$ (e.g., UCLID5); a parent language, $P$ (e.g., Python); a child language, $C \subset P$ (e.g., a subset of Python); and a compiler, $f$, from $C$ to $T$ (e.g., a syntax-directed translation from the subset of Python to UCLID5). Given an input task, we create a prompt $q$ that asks the LLM $L$ to generate code, $p \in C$, which satisfies the task description, $q$. The second step of the workflow is to repair $p$. If there is nothing wrong with $p$, or $p$ can be fixed using formal techniques described in §5.3 and §5.4, then repairing will generate a new, complete program $p'$ and return $f(p')$ (i.e., a program in the target language, like Fig. 1b). Frequently, however, repairing will generate a partial program containing "holes" (denoted "??"). For example, Fig. 5b shows the first $p$ generated for the task in Fig. 1a and Fig. 5a shows the corresponding partial program $p'$ that was automatically generated using our formal techniques. Programs with holes cannot be compiled to the target language, so the third step of the workflow is ask the LLM to complete the partial program $p'$, generating a new $p$. We use the template in Fig. 4b to generate the LLM prompt. Fig. 6a shows the output generated by gpt-3.5-turbo-0125 when asked to repair the partial program in Fig. 5a. This program is still incorrect, but, it is now close enough that we can automatically repair it to a complete program without holes. Fig. 6b shows the relevant portion of that complete program. This final, complete program is directly translated to the output in Fig. 1b. §5 elaborates on each of the workflow components.

To understand the subtle mistake in Fig. 1b one needs some understanding of UCLID5 [34, 37]. UCLID5 is a *verification language* used to model hardware and software systems as "state machine" like transition systems and to automatically check if the transition system does indeed satisfy a formal logic specification. UCLID5 transition systems primarily consist of a state space given by a set of variable declarations (e.g., lines 2-4 in Fig. 1b), an initialization block that represents a set of acceptable start states (e.g., lines 5-7 in Fig. 1b), and a transition relation block that defines how the transition system moves from one state to the next state (e.g., code starting at line 8 in Fig. 1b). The `var` keyword is used to declare variables that are internal to the transition system in question while the `input` keyword is used to declare read-only variables that are outside the control of the transition system in question. Fig. 1b passes all compiler checks but has a subtle semantic mistake on line 4: `var pedestrian:  boolean;` should be `input pedestrian:  boolean;` because the presence of a pedestrian should not be controlled by the traffic light transition system. When manually assessing correctness in §6, this subtle mistake would prevent us from marking this example as fully correct.

## 4  Background

In this section we provide the necessary technical background to understand the formal techniques that are used in our approach and are described in §5.3 and §5.4.

### 4.1  Algebraic Data Types and Abstract Syntax Trees

Algebraic data types (ADTs) are a representation of finite trees that are common in functional programming languages. We provide an informal definition of ADTs and point the interested reader to Barrett et al. [6] for a more formal treatment.

An ADT consists of a set of constructors (node names), selectors (directed edge names), and testers (predicates). Each constructor has a fixed set of selectors associated with it (possible edges out).

```
from FormalVerificationLibrary import Module # GTCODE 1: imports a class from the hypothetical library
m = Module("myModule") # GTCODE 2: creates an instance of an imported class
print(m) # GTCODE 3: uses a dunder method (in this case __str__) of an imported class
```

Figure 3: Hypothetical LLM output with grounded theory codes as comments. Codes are determined and assigned manually.

Each selector has an associated type (each edge can only point to a fixed set of node names). Every constructor is associated with exactly one unique tester: that tester returns true on a given tree *iff* the root of the tree is labeled with the corresponding constructor. Every instance of an ADT (a particular finite tree built from those node and edge names) must be acyclic.

Abstract syntax trees (ASTs) are instances of ADTs—i.e., ASTs are concrete finite trees—that represent programs. Every programming language has a corresponding ADT that represents a superset of all possible programs in that programming language. Some instances of a languages's ADT will not correspond to valid programs, e.g., if they do not additionally type check.

### 4.2 Satisfiability Modulo Theories and Weighted Maximum Satisfiability

Satisfiability Modulo Theories (SMT) [7] is a class of problems generalizing Boolean satisfiability (SAT) to first-order logics with additional background logical theories. We give an informal presentation of satisfiability modulo theories (SMT) that focuses on only the theory of ADTs and covers weighted maximum satisfiability (MAX-SMT). Further details may be found in a book chapter [7].

Let $\Gamma$ be a set of ADTs and let $V$ be a set of typed variables (pairs of names and types). For simplicity we assume that variable types are exactly elements of $\Gamma$. In reality, variables can also have function types (e.g., $V \doteq \{(z, \texttt{Bool}), (f, \texttt{Bool} \mapsto \texttt{Bool})\}$ would be fine). An *atomic formula* is an equation or the application of a single tester over $V$ (e.g., $z = \texttt{True}$ and $\texttt{is\_And}(z)$ are both atomic formulas). A *theory literal* is an atomic formula or its negation. A *clause* is a set of theory literals. A *conjunctive normal form (CNF) formula*, or *formula* for short, is a set of clauses. For example, if $\Gamma \doteq \{\texttt{Bool}\}$ and $V \doteq \{(z, \texttt{Bool})\}$, then $\{\{z = \texttt{True}\}, \{\texttt{is\_And}(z)\}\}$ is a formula. An *interpretation* is a mapping from variables to elements of their corresponding type. For example, $I(x) \doteq \texttt{True}$ if $x = z$ and $I(x) \doteq \texttt{False}$ otherwise, is an interpretation. Interpretations are extended to atomic formulas in the natural way When an atomic formula $\phi$ evaluates to true under an interpretation $I$, we say that $I$ *satisfies* $\phi$ and write $I \models \phi$. We extend the notion of satisfiability to literals, clauses, and formulas in the natural way and reuse the same notation. The satisfiability modulo theories problem is to determine if, for a given formula $\phi$, there exists an interpretation $I$ such that $I \models \phi$. When such an $I$ exists we say that $\phi$ is *satisfiable* (**sat**). When no such $I$ exists, we say that $\phi$ is *unsatisfiable* (**unsat**). For example, $\{\{z = \texttt{True}\}, \{\texttt{is\_And}(z)\}\}$ is **unsat**.

The maximum satisfiability problem is, for a given (CNF) formula $\phi$, to determine the largest subset of $\phi$ that is **sat** (solvers aim to satisfy as many clauses as possible). The weighted maximum satisfiability problem (MAX-SMT) is a variation with two differences. First, some clauses can be "hard"—meaning they must be satisfied. Second, every "soft" clause (any clause that is not "hard") has an associated weight. The problem is then to determine subset of $\phi$ that maximizes the sum of weights while being **sat** and containing all "hard" clauses.

## 5 Approach

In this section we describe the SPEAC approach. We begin with how to select $P$ and $C$ (§5.1). We then describe the promoting we use to interface with LLMs (§5.2) and two formal techniques at the heart of our automated repair (§5.3 and §5.4).

### 5.1 Synthetic Programming Elicitation

In this section, we present *synthetic programming elicitation* as a kind of programming study—where LLMs are the subject—that follows three steps. The results of these studies are used to select $P$ and $C$. We call this process synthetic programming elicitation since it is inspired heavily by natural

programming elicitation [29]. To make this section more concrete, for each step of the study, we describe the specific steps we followed for targeting UCLID5.

**First Step: Setup.** First prepare LLM subjects, select a target language, and collect or construct a set of programming tasks. In our case, our target language is UCLID5, our LLM subjects are gpt-4-turbo-2024-04-09 and gpt-3.5-turbo-0125, and our programming tasks are a set of regression tests written by UCLID5 developers with corresponding natural language descriptions that were written by hand (see §6 for more details on all the benchmarks).

The second part of the study setup is to create multiple kinds of specialized prompts. The first will ask for the output to be in the target language directly. Subsequent prompts will ask for the output to use an imaginary API in a popular host language, like Python or Java. For example, for UCLID5, we generated prompts that look like the following.

1. "Write a UCLID5 model for a system that counts the number of times the temperature exceeds a threshold [. . .]"

2. " FormalVerificationLibrary is a Python library for generating verification models [. . .]. Use the FormalVerificationLibrary to model a system that counts the number of times the temperature exceeds a threshold [. . .]"

**Second Step: Execute.** Now we execute every LLM subject on every prompt and collect the outputs. Each query should be executed independently.

**Third Step: Analyze and Design.** Finally, we analyze the collected LLM outputs and select $P$ and $C$ based on the analysis. To do this analysis, we follow a lightweight version of grounded theory [19]. See e.g., Stol et al. [39] for general guidelines on how to use grounded theory in software engineering research. See e.g., Barke et al. [5] for a specific, detailed case-study. Synthetic programming elicitation studies are inspired by and borrow many of the methods from those works. Our studies are categorically different, however, because we are not working with humans. We cannot interact with our participants in the same way, e.g., we cannot conduct semi-structured interviews. The goal of our studies is to design an intermediate language that more closely matches the code that LLMs tend to generate when compared to our desired target language.

The first step of our lightweight grounded theory analysis is to "code" the outputs generated by the LLMs. In grounded theory parlance, "code" is akin to manually labeling parts of the generated outputs. For example, Fig. 3 shows a hypothetical LLM output for our running example along with grounded theory codes as comments. The second step is to group codes into concepts. Concepts are slightly more abstract than codes. For example, a concept that may emerge from Fig. 3 is the group of codes 1, 2, and 3. The corresponding concept could be "using a class from the hypothetical library." In the third step, we group concepts into categories. For example, we may group concepts related to the object oriented programming paradigm as one category. Finally, we select a $P$ and $C$ that are consistent with the final categories we observed across multiple prompts.

In our study, we found that, when asked to write UCLID5 code without any special prompting or training, no LLM was able to produce code that parses (660 attempts: 10 samples per LLM per benchmark, 33 benchmarks, two LLMs). Worse still, the code generated by LLMs was inconsistent, with each LLM giving different outputs that resemble different programming languages at different times. When asked to write Python code that used a non-existent formal verification API, however, the LLM outputs were more consistent. Therefore, we selected Python as our parent language, $P$.

Specifically, the Python code was more consistent because LLM outputs fell into three broad categories, which we call "from-scratch," "procedural," and "object-oriented," respectively. Programs in the "from-scratch" category used existing APIs (e.g., the Z3 solver API [15]) to re-implement what UCLID5 does, e.g., to manually model transition systems. This was the smallest significant category. Programs in the "procedural" category imported a class from the hypothetical API, created an instance of it, and then called methods from the class to declare variables, assert specifications and so on. Programs in the "object-oriented" category imported a class from the hypothetical API and extended it, including methods that correspond closely to parts of UCLID5 code. For example, these extended classes often included a method that represented a transition relation—a critical component of UCLID5 programs and of transition systems generally.

After analyzing the outputs, we concluded that it would be easiest to translate Python programs from the "object-oriented" category to UCLID5. For example, we observed that methods which represent

| | |
|---|---|
| 1 Write [PARENT LANGUAGE] code to complete the following task. | Fix the following [PARENT LANGUAGE] code by replacing every occurrence of '??' with the correct code. 1 |
| 2 > [TASK GOES HERE] | |
| 3 Reply with your code inside one unique code block | [CODE WITH HOLES GOES HERE] 2 |
| 4 [DESCRIBE CHILD LANGUAGE] | Make sure that your code completes the following 3 |
| 5 I can definitely do that! Here is the code: | task. |
| 6 ``` | [LINES 2–6 OF (a)] 4 |

(a) Template for prompt $q$ in Fig. 2         (b) Template for prompt $q'$ in Fig. 2

Figure 4: Partial SPEAC prompt templates for first generation (a) and hole filling (b).

transition relations used a limited set of names, and that methods themselves could be compiled to UCLID5 rather directly. Therefore we defined a subset of Python that matches the "object-oriented" category and used that as our child language, $C$. Essentially, $C$ is an abstract class that the LLM must extend. For example, the abstract class includes an abstract method called "next" that corresponds to the transition relation. Fig. 6b shows a portion of an example of a Python program in $C$ and Fig. 1b shows a portion of its corresponding UCLID5 program.

## 5.2 (Minimal) Prompting

After our synthetic programming elicitation study—once $P$ and $C$ have been selected—we use minimal prompting to interface with LLMs. For example, for UCLID5, we use the prompt template in Fig 4a to create initial programs and the prompt template in Fig. 4b to fill in holes. More advanced prompting techniques are complementary and could help here. However, for this work, we used minimal prompting so as to independently evaluate our contribution.

## 5.3 Largest Consistent Subtree

Given a program $p$ in $P$, but not in $C$, our aim is to remove as little as possible from $p$ to bring it into the language $C$. That is, given the AST for $p$, we wish to generate the largest subtree of the AST, possibly containing holes, that is not rejected by the static checks of the language $C$. For example, the code in Fig. 5b contains some errors, including mismatches between variable and constant types (UCLID5 does not automatically cast constants and so the line self.count = 1 is interpreted as assigning the integer literal 1 to the bitvector variable self.count, which is a type error).

We find the largest consistent subtree in three steps. First, we use an error tolerant parser to build an AST, $A$, for the language $P$. Second, beginning at the root of $A$, we recursively descend and prune anything that cannot exist in an AST for the language $C$, placing holes wherever they are needed. This is an aggressive program slice, similar to Akhundov et al. [2], who give a new AST $A'$. $A'$ may or may not pass compiler checks, like type checking, or be semantically correct.

The third step finds the minimal number of AST nodes that need to be replaced with holes such that all static checks of the language $C$ pass, using the approach of Pavlinovic et al. [32]. Specifically, for every static check of the language, for every node of the AST, we generate a set of soft clauses. Let $S$ be the the union of all the generated clauses, and let $S'$ be the largest subset of clauses that is **sat** (the output of a MAX-SMT solver). Any clause, $c$, that appears in $S$ but does not appear in $S'$ represents a bug: if we add $c$ to $S'$, then we will get an **unsat** set of clauses. So, for every such $c$, we replace the AST node that generated $c$ with a hole.

For example, in UCLID5, the two statements var x: bv32; and x = 0; would produce a set of clauses corresponding to the assertions that (a) x is a bitvector and that (b) x is an integer (since 0 is an integer). These assertions, together, are **unsat**, so MAX-SMT would return a set of clauses that does not include both (a) and (b). One solution would be to remove the bitvector type, producing the partial, but consistent, two statements var x: ??; and x = 0;.

We can bias the solver to include one or the other by adjusting the weights on the corresponding clauses. In practice, we assign clauses weights that are proportional to the depth of the corresponding node in the AST. This ensures that edits are closer to the leafs of the AST but it breaks syntactic minimality guarantees. In the case where weights are uniform and every node corresponds to exactly one clause, the MAX-SMT approach will make the minimal edit to the AST.

```
1   class TrafficLight(Module):              24  def next(self):                        class TrafficLight(Module):        1
2       def types(self):                     25      if self.state == 0:                    def locals(self):              2
3           self.state_t = BitVector(2)      26          self.sigG = False                      self.state = int           3
4       def locals(self):                    27          self.sigY = False                      self.count = int           4
5           self.state = BitVector(2)        28          self.sigR = True                       self.pedestrian = bool     5
6           self.count = BitVector(6)        29          self.count = 0                     def outputs(self):             6
7           self.pedestrian = Boolean()      30          if self.count < 60:                    self.sigG = bool           7
8           self.sigG = Boolean()            31              self.count += 1                    self.sigY = bool           8
9           self.sigY = Boolean()            32          else:                                  self.sigR = bool           9
10          self.sigR = Boolean()            33              self.state = 1                 def init(self):               10
11      def inputs(self):                    34      ...                                        self.state = ??           11
12          self.pedestrian = Boolean()      35      elif self.state == 3:                      self.count = ??           12
13      def outputs(self):                   36          self.sigG = False                      self.pedestrian = ??      13
14          self.sigG = Boolean()            37          self.sigY = False                      self.sigG = ??            14
15          self.sigY = Boolean()            38          self.sigR = False                      self.sigY = ??            15
16          self.sigR = Boolean()            39          if self.count < 60:                    self.sigR = ??            16
17      def init(self):                      40              self.count += 1               def next(self):               17
18          self.state = BitVector(2)        41          else:                                  if self.state == 0:       18
19          self.count = BitVector(6)        42              self.state = 0                         self.sigG = False     19
20          self.pedestrian = Boolean()                                                            self.sigY = False     20
21          self.sigG = Boolean()                                                                  self.sigR = True      21
22          self.sigY = Boolean()                                                                  self.count = 0        22
23          self.sigR = Boolean()                                                              ...                       23
```

(a)                                                                                       (b)

Figure 5: Partial response for task in Fig. 1a using gpt-3.5-turbo-0125 (a) and partial first repair (b). Line 5 in (a) declares `state` as a local variable of bit-vector type; lines 25, 33, and 42 use `state` as an integer. Line 3 in (b) repairs the type of `state` to be an integer.

## 5.4 Model-Based Repair

Once we have a satisfiable set of clauses, we can generate a corresponding satisfying model and use the model to repair the partial program. This is where our work most differs from Pavlinovic et al. [32]. For example, the partial program `var x:   ??;` from above would correspond to the set of clauses `is_Integer(??)`. This clause would be satisfied by setting `??` to `integer` and so we can repair the partial program to be `var x:   integer;`.

Fig. 6 shows one buggy AST (Fig.6a) and the corresponding fix (Fig. 6b). For example, the variable `count` is declared as an integer in the repaired program because it is used as an integer in the buggy program. For repairs that cannot be automatically resolved by the MAX-SMT model, we use the LLM to generate code to replace the AST holes, as shown in Fig. 2.

## 6 Evaluation

In this section, we implement a prototype of SPEAC for UCLID5, called Eudoxus, and use it to evaluate the performance of SPEAC across three research questions. RQ1: Does Eudoxus perform better than LLM baselines on syntactic correctness (passing compiler checks)? RQ2: Does Eudoxus perform better on semantic correctness? RQ3: Does MAX-SMT cost too much computation time? Eudoxus uses the $P$ and $C$ described in §5 and is implemented in Python.[2] We use tree-sitter to parse and search partial programs, and Z3 [15, 9] to solve MAX-SMT queries. We allow Eudoxus to repeat the repair loop five times. All MAX-SMT queries are solved locally on a 2.3 GHz Quad-Core Intel Core i7 with 32 GB of RAM. All LLM calls are made through the OpenAI Python API.

We compare Eudoxus against three LLM baselines: few-shot, self-repair, and fine-tuning. All using gpt-3.5-turbo-0125 (GPT3t) and gpt-4-turbo-2024-04-09 (GPT4t). **Few-shot with and without Chain-of-Thought.** For the few-shot baseline, we provide examples taken from the UCLID5 GitHub repository to the LLM in context. We tried with one in context example, and then again with three in context examples. We also combine few-shot prompting with chain-of-thought [44]. **Fine-tuning.**

---

[2]Available at: `https://github.com/FedericoAureliano/eudoxus`

```
 1  class TrafficLight(Module):              class TrafficLight(Module):               1
 2      def locals(self):                        def locals(self):                     2
 3          self.state = 0                           self.count = int                  3
 4          self.count = 0                           self.pedestrian = bool            4
 5          self.pedestrian = False                  self.sigG = bool                  5
 6      def outputs(self):                           self.sigR = bool                  6
 7          self.sigG = False                        self.sigY = bool                  7
 8          self.sigY = False                        self.state = int                  8
 9          self.sigR = False                                                          9
10      def init(self):                          def init(self):                      10
11          self.state = 0                           self.state = 0                   11
12          self.count = 0                           self.count = 0                   12
13          self.pedestrian = False                  self.pedestrian = False          13
14          self.sigG = False                        self.sigG = False                14
15          self.sigY = False                        self.sigY = False                15
16          self.sigR = True                         self.sigR = True                 16
17          ...                                      ...                              17
```

(a)                                          (b)

Figure 6: Partial response for Fig. 5a using gpt-3.5-turbo-0125 (a) and partial second repair (b). The `outputs` method should declare variables and their types but on lines 6-9 of (a) it instead assigns variables to values (i.e., `False` instead of `bool`). The repair in (b) declares these variables with the appropriate types on lines 5-7.

As with many VLPL, there are very limited number of natural language to UCLID5 examples in the public domain, and certainly not enough for fine-tuning. We do, however, have access to 317 regression tests from the open-source code base. For the purposes of fine-tuning, we use self-instruct [43] to first ask the model to summarize the UCLID5 regression tests in natural language and then provide this as the natural language description during fine-tuning. We fine-tune GPT3t for 284 steps with learning-rate multiplier 0.16. **Self-repair.** We gave the LLMs compiler feedback in a self-repair loop starting from a zero-shot prompt. We use five iterations to be consistent with Eudoxus.

We use three sets of UCLID5 benchmarks. The first set is a curated set of 33 regression tests with handwritten natural language descriptions. These tests are designed to cover a broad range of UCLID5 features and were used for the synthetic programming elicitation study in § 5.1. The second set is a set of 317 regression tests without descriptions taken directly from the UCLID5 GitHub repository. These tests were used for fine-tuning our baseline model, as described above. This set is representative of the quantity and quality of data that a user may have available for a VLPL. The third and final set is a set of 33 exercises and examples taken from three textbooks [4, 24, 21]. These benchmarks cover formal modeling of concurrent systems, linear-time properties, model checking and systems with discrete dynamics. We used this final set for the end-to-end evaluation below.

## 6.1 Results

We run two variations of Eudoxus (one using GPT3t, one using GPT4t) and 11 variations of the baselines on all 33 textbook benchmarks. We report the fraction of outputs that pass all compiler checks ("parse," for short) and semantic correctness over all 11 approaches in Table 1. Semantic correctness is manually assessed by one author using a combination of manual reasoning and hand-written specifications for all programs that compile. Correctness is rated on a scale from from $1 - 5$, where $1$ is entirely wrong, and $4$ indicates that the model is correct with only a few minor errors (e.g., the example output described in §3 and Fig. 5). Intuitively, any score of $\geq 4$ indicates that we believe the output would be useful to text-to-code users.

**RQ1: Syntactic Correctness.** Eudoxus outperforms all baselines in terms of compiler checks (see "Parse Rate" in Table 1), passing all compiler checks on $78\%$ of benchmarks. There are four benchmarks on which Eudoxus hits the iteration limit and fails to produce a result, and three benchmarks with small syntax errors due to bugs in our Eudoxus implementation (e.g., failing to extract code snippets from LLM outputs or printing UCLID5 code incorrectly). In contrast, we find that GPT3t and GPT4t rarely produce UCLID5 models that even parse. The best results for GPT3t

| | Parse Rate | Semantic Score | | | | |
|---|---|---|---|---|---|---|
| | | 1 | 2 | 3 | 4 | 5 |
| Eudoxus (GPT4t) | 24/33 | 1/24 | 1/24 | 3/24 | 8/24 | 11/24 |
| Eudoxus (GPT3t) | 28/33 | 3/28 | 6/28 | 5/28 | 5/28 | 9/28 |
| Fine-tuned GPT3t | 2/33 | 0 | 2/2 | 0 | 0 | 0 |
| One-shot with COT (GPT4t) | 1/33 | 0 | 0 | 0 | 1/1 | 0 |
| One-shot with COT (GPT3t) | 0 | - | - | - | - | - |
| One-shot (GPT4t) | 0 | - | - | - | - | - |
| One-shot (GPT3t) | 0 | - | - | - | - | - |
| Three-shot with COT (GPT4t) | 3/33 | 0 | 0 | 0 | 2/3 | 1/3 |
| Three-shot with COT (GPT3t) | 1/33 | 0 | 0 | 0 | 0 | 1/1 |
| Three-shot (GPT4t) | 4/33 | 0 | 0 | 0 | 3/4 | 1/4 |
| Three-shot (GPT3t) | 2/33 | 0 | 0 | 1/2 | 0 | 1/2 |
| Self-repair (GPT4t) | 0 | - | - | - | - | - |
| Self-repair (GPT3t) | 0 | - | - | - | - | - |

Table 1: Eudoxus compared to baselines. We report the semantic score over all correctly parsed models. 1 is completely wrong; 5 is fully correct. Eudoxus is limited to five LLM calls per benchmark, and four benchmarks hit this limit.

come from fine-tuning, but it is only able to produce two programs that parse. The best results for GPT4t come from three-shot prompting, but it is only able to produce four programs that parse. Given this data, we answer RQ1 in the affirmative: Eudoxus perform better than standard LLM baselines on syntactic correctness. Even more interesting, the two versions of Eudoxus generated programs that passed all compiler checks for 30/33 unique problems; the 11 versions of the baselines together generated programs that passed all compiler checks for only 6/33 unique problems.

**RQ2: Semantic Correctness.** Every unique problem that the baselines perform well on—four unique problems where some baseline scored four or higher—was also solved by Eudoxus. These are likely the easiest problems in the test set. On the other hand, $33/52$ programs (24 unique problems) that pass compiler checks produced by Eudoxus scored four or higher. This data suggests that our approach does not negatively impact semantic performance, but it is difficult to draw conclusions since so few of the programs generated by the baselines pass compiler checks.

**RQ3: MAX-SMT Performance.** In terms of execution time, Eudoxus with GPT3t took an average of 0.9 seconds (SD 0.6) in repair steps and 7.2 seconds (SD 4.6) in LLM calls. Eudoxus with GPT4t took an average of 1.8 seconds (SD 2.0) in repair steps and 35.1 seconds (SD 24.7) in LLM calls. We conclude that, in terms of execution time, LLM calls are more expensive than our static repairs.

### 6.2 Threats to Validity

While our results are promising, there are three limitations to our evaluation. First, we only evaluated SPEAC on one VLPL, UCLID5. It remains to be seen if our results can be replicated across different VLPLs, especially those in different domains. Second, we only evaluated SPEAC with two LLMs. It is possible that our work will not generalize across different kinds of LLMs. Third, only one author evaluated the semantic correctness of programs generated by Eudoxus. While the main takeaway of our work is that we are able to generate syntactically correct programs where baselines cannot, it is possible that we have over or under-estimated semantic correctness.

## 7 Conclusion

We have presented a synthetic program elicitation and compilation method (SPEAC) that supports natural language to code generation for very low resource programming languages (VLPLs). The two key ideas behind SPEAC are (1) to design an interface that is "natural" for LLM "users" and (2) to use deductive techniques, which could be deemed too aggressive for human users, to automatically repair LLM outputs when possible. We implemented a prototype of SPEAC called Eudoxus that targets the UCLID5 VLPL and evaluated it on a set of 33 benchmarks from textbooks in the same domain as UCLID5. Eudoxus performs significantly better than standard LLM baselines on syntactic correctness without sacrificing semantic correctness.

## Acknowledgments and Disclosure of Funding

We would like to thank Adwait Godbole, Ameesh Shah, Max Willsey, and the anonymous reviewers for their insightful feedback. We would like to thank Justin Lubin for pointing us to natural programming elicitation. Part of this work was done during the Transfer-to-Excellence Summer Research Program at UC Berkeley in 2023 and part during UC Berkeley's CS 294-260: Declarative Program Analysis and Optimization in 2024. This work was supported in part by a Qualcomm Innovation Fellowship, a Royal Academy of Engineering Research Fellowship, DARPA Contract FA8750-23-C-0080 (ANSR), C3DTI, an Amazon Research Award, NSF grant 2303564, by Nissan and Toyota under the iCyPhy center, and by Intel under the Scalable Assurance program.

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
