# OpenReview forum: "Synthetic Programming Elicitation for Text-to-Code in Very Low-Resource Programming and Formal Languages"
_NeurIPS.cc/2024/Conference — NeurIPS 2024 poster_

### Official Review · Reviewer_6oM2 · 2024-06-24

**Soundness:** 2
**Presentation:** 3
**Contribution:** 3
**Rating:** 7
**Confidence:** 4

**Summary:**

Recent advances in large language models (LLMs) for code applications have demonstrated remarkable zero-shot fluency and instruction following on challenging code-related tasks, ranging from test case generation to self-repair. However, the authors note that these models struggle to compose syntactically valid programs in programming languages that were unrepresented during pre-training, known as very low-resource programming languages (VLPLs). VLPLs are critical in various settings, including domain-specific languages for internal tools and tool-chains, and legacy languages. Inspired by program elicitation, the authors propose creating a hallucinated library within a high-resource language that can be automatically compiled to the VLPL. This library enables the LLM to generate and self-repair code within the syntax of a familiar language. Specifically, the authors introduce Synthetic Programming Elicitation and Compilation (SPEAC), an approach that enables LLMs to generate syntactically valid code even for VLPLs. The authors empirically evaluate the performance of SPEAC in a case study and find that, compared to existing retrieval and fine-tuning baselines, SPEAC produces syntactically correct programs more frequently without sacrificing semantic correctness.

**Strengths:**

+ Important area
+ Novel idea
+ Good performance

**Weaknesses:**

- Missing some details
- Fail to consider code generation approaches via self-repair or self-debugging
- Only conducted experiments on UCLID5

**Questions:**

Overall, I appreciate the idea presented in this paper.

The research tackles a critical problem in the field of LLMs for code generation, particularly focusing on VLPLs, which are often overlooked but essential in many applications. The concept of using MAX-SAT to bridge the gap between high-resource and very low-resource languages is innovative and offers a new direction for future research. The empirical results demonstrate that SPEAC outperforms existing baselines in producing syntactically correct programs without losing semantic accuracy.

However, the authors need to address some issues.

1. **Missing some details**:

The authors omit important details about the semantic score. How is the semantic score computed? Is it done automatically or labeled manually? If labeled manually, how many people were involved, and what is the kappa score?

2. **Fail to consider code generation approaches via self-repair or self-debugging**:

While the proposed SPEAC approach is promising, the paper does not adequately address how it compares to or integrates with existing methods that leverage self-repair or self-debugging capabilities in LLMs. These approaches are also relevant to this paper.

3. **Only conducted experiments on UCLID5**:

The evaluation of SPEAC is limited to a single case study on UCLID5. Broader validation across multiple VLPLs would enhance the generalizability of the findings.

**Limitations:**

See Questions, thanks.

---

> ### Author Rebuttal · Authors · 2024-08-07
>
> We thank the reviewer for their thoughtful feedback. Please see below for our answers to the specific questions posed.
>
> **6oM2-Q1: Details of the semantic score?**
>
> The semantic scores were manually judged by one professor with experience teaching formal methods (undergraduate and graduate level courses). This small number of judges is definitely a limitation and we will make this explicit in the paper. However, the key takeaway from the results is that Eudoxus lifts the syntactic correctness from 2/33 (Fine-tuned GPT-3.5-turbo) to 28/33 (Eudoxus using GPT-3.5-turbo) while not terribly harming semantic correctness.
>
> **6oM2-Q2: Self-repair or self-debugging?**
>
> Please see response to reviewer 9eZc question 3 (9eZc-Q3).
>
> **6oM2-Q3: Only UCLID5?**
>
> The goal of the UCLID5 case study was to conduct an in depth demonstration of our techniques for a very low resource language that looks very different from high resource languages and has a wide range of features. UCLID5 supports software verification, like Boogie (Barnett et al. 2005); hardware verification, like ABC (Brayton and Mishchenko 2010); and can even be used for modeling distributed systems, like P (Desai et al. 2013). We acknowledged that including only one case study is a threat to validity (lines 328-331) but UCLID5’s wide range of features is evidence for the generalizability of our approach.

---

> ### Comment · Reviewer_6oM2 · 2024-08-12
>
> Thanks. I like the idea of this paper and have decided to give it a score of 7. I was initially wavering between a 6 and a 7 because while the idea of this paper is really good, the number of human judges is too small. Ultimately, I settled on a 7.

---

### Official Review · Reviewer_9mNj · 2024-07-08

**Soundness:** 2
**Presentation:** 1
**Contribution:** 2
**Rating:** 4
**Confidence:** 4

**Summary:**

In this paper, the authors present a novel approach called SPEAC (Synthetic Programming Elicitation and Compilation) to enable large language models (LLMs) to generate syntactically valid code for very low-resource programming languages (VLPLs). The approach involves creating a hallucinated library within a high-resource language, which can be compiled into the VLPL. The paper includes a case study demonstrating that SPEAC outperforms existing methods in producing syntactically correct programs. Evaluation results demonstrate that SPEAC outperforms all baselines on syntactic correctness (see “Parse Rate” in Tab 1), achieving full syntactic correctness on 78% of benchmarks.

**Strengths:**

The paper addresses a novel problem of generating code for VLPLs by leveraging LLMs, which is a relatively unexplored area.

The empirical evaluation demonstrates the effectiveness of SPEAC compared to other baselines.

**Weaknesses:**

The claims made about the significant improvements achieved by SPEAC are not fully supported by a diverse set of experiments.

The paper is poorly organized and lacks a clear, logical flow.

**Questions:**

The first paragraph in Section 2 is very disorganized. If the authors aim to discuss LLMs for code generation, they should focus on them rather than the benchmarks. It would be beneficial to include more examples of LLMs and either remove the benchmarks discussion or introduce benchmarks in a separate section dedicated to the code generation area.

The experimental section is weak. Reviewers can only find the experiment results on the eighth page. The authors should present the experimental setup, results, and analysis more prominently and earlier in the paper to support their claims better.

---

> ### Author Rebuttal · Authors · 2024-08-07
>
> We thank the reviewer for their thoughtful feedback. Please see below for our answers to the specific questions posed.
>
> **9mNj-Q1: Reorganization of the paper?**
>
> We are happy to make all the suggested organization changes under the questions header. Specifically, we are happy to (1) reorganize the first paragraph of Section 2; (2) introduce benchmarks in a separate section; and (3) present the experimental setup, results, and analysis more prominently and earlier in the paper. We estimate these changes should take no more than one day of work.
>
> **9mNj-Q1: More LLMs?**
>
> We are happy to include more examples of LLMs, as suggested, although our initial experimentation suggests that their performance generating syntactically correct UCLID5 code will be comparable to the OpenAI models (i.e., poor). If there are specific LLMs the reviewer believes we should add, we are happy to follow suggestions.

---

> > ### Comment · Reviewer_9mNj · 2024-08-14
> >
> > Thank you for the responses, and sorry for the late response due to a flurry of proposals and review dues. I have read other reviewer's rebuttals and the author's response. I have increased my overall assessment as some of my concerns have been addressed. While others are shown in below.
> >
> > **Other LLMs** I recommend conducting experiments in OpenCodeInterpreter, DeepSeek-Coder, XwinCoder, CodeLlama, WizardCoder, and Starcoder2 in open-source LLMs.

---

### Official Review · Reviewer_9eZc · 2024-07-13

**Soundness:** 3
**Presentation:** 3
**Contribution:** 2
**Rating:** 5
**Confidence:** 2

**Summary:**

The paper presents a framework to generate programs given natural language and targeting very low-resource programming languages. They first choose a language well-represented in the training data (in this case Python), and then make the LLM generate in a subset of that language, and then compile the generated program to the target language. When the generated program is checked by formal techniques and found to be inconsistent, some parts of the program are replaced with holes and fixed with LLM again. They apply this method to a very low-resource language VLPL and show that it is able to generate programs that parse (24 out of 33 test problems) and 11 out of 24 have fully correct semantic meaning, while the baseline few-shot prompting and fine-tune methods obtain near zero parsable programs.

**Strengths:**

* The results are very strong compared to the baseline few-shot prompting and fine-tune methods, which obtain near zero parsable programs.
* The framework is described in a general way and potentially applicable to other low-resource programming languages as well.
* The system integrating LLM program generation and formal methods for checking the program is very interesting and demonstrates the effectiveness of the method empirically.

**Weaknesses:**

A major concern of the paper is that there should be other stronger and popular methods to use as baselines. For example, a common method is iterative fixing of unparsable code by prompting LLM with the parser feedback. Also, many-shot examples (more than one) and including language specification in the prompt may also help, which is also a very standard prompting technique among the community. The method presented here would likely still work better, but it would be more informative to compare with those methods and see how much improvement is made.

**Questions:**

* Would few-shot prompting baseline methods with more than one example work better?
* How does constraint decoding compare to the method here to get a parsable program in VLPL instead of Python subset?
* How does the method compare with reflexion-style methods where iterative prompting is used to fix the problem by including feedback from the parser in the prompt at each iteration?
* It is mentioned that there is a set of regression tests comprised of 317 tests taken directly from the UCLID5 GitHub repository. How are these tests used? Are they used to evaluate besides the results on the 33 test problems in Table 1?
* How does the self-instruct with fine-tune method work specifically? How many synthetic programs are generated?
* Can you provide the full prompt that is used for generation and fixing? Is the code in the appendix A included in the prompt in addition to the prompt template shown in Fig 4?

**Limitations:**

Limitations are addressed.

---

> ### Author Rebuttal · Authors · 2024-08-07
>
> We thank the reviewer for their thoughtful feedback. Please see below for our answers to the specific questions posed.
>
> **9eZc-Q1: Would few shot prompting work better?**
>
> Adding more examples improves both the baseline’s performance and Eudoxus’ performance (which currently uses no examples, see Fig. 4). Using few-shot with 3 in-context examples leaves gpt-3.5-turbo's performance with zero successful compilations. For gpt-4-turbo, we find both with and without COT produces successfully compilation for 3/33.
>
> **9eZc-Q2: Comparison with constrained decoding?**
>
> Our work is complementary to constrained decoding in two ways. First, one of the pitfalls of constrained decoding is that it will perform poorly if the external constraints do not align with the underlying vocabulary of the LLM (Beurer-Kellner et al. 2024), i.e., if the desired output is “unnatural” to the LLM then constrained decoding does poorly. Synthetic programming elicitation could help bridge this gap for very low resource programming languages.
>
> Second, most existing constrained decoding techniques support only context-free languages (Beurer-Kellner et al, 2024) but most programming languages are at-least context-sensitive languages. Our MaxSMT driven repair can be applied after constrained decoding to fix more complicated, context-sensitive errors, like type-checking bugs.
>
> **9eZc-Q3: Comparison with reflection style methods?**
>
> We gave gpt3.5 and gpt4 compiler feedback in a self-repair loop starting from a zero-shot prompt and found no benefits. Specifically, our repair prompts provided the LLMs with their previously generated code and the corresponding compiler error. We let both LLMs iterate up to five times (same as Eudoxus in our empirical evaluation) but neither LLM was able to successfully repair its own code. We believe this is because the baseline LLM outputs are too far from correct outputs: there are just too many errors to fix. This is consistent with observations from existing work for verification languages, where self-repair alone shows only modest improvements (e.g., Loughridge et al. 2024).
>
> Reflection style methods may be more impactful when combined with more advanced techniques (as opposed to a zero-shot starting point), like Eudoxus. Our LLM repair step is currently extremely simple, giving no compiler feedback and doing no self-debugging (see Fig. 4b). Combining our work with complementary reflection style methods is a promising avenue for future work.
>
> **9eZc-Q4: How are the regression tests used?**
>
> The 317 are used for fine-tuning and for synthetic programming elicitation. See 9eZc-Q5 for more information on fine-tuning. For synthetic programming elicitation, we wrote natural language descriptions for a subset of the regression tests and used these natural language descriptions (without the corresponding regression test) to study the “natural” behavior of the LLMs. We mention this on lines 199-200 but will make it more clear.
>
> **9eZc-Q5: How does self-instruct with fine-tuning work?**
>
> For self instruct, we took existing tests in UCLID5 and asked the off-the-shelf LLM to create a plausible natural language description. This provides us 317 training pairs of natural language and valid UCLID5 on which to fine-tune. Just as in the self-instruct paper (Wang et al. 2023), the natural language descriptions (questions) are synthetic but the programs (solutions) are not synthetic. We briefly describe this on lines 295-297 but will clarify and add the appropriate citation.
>
> **9eZc-Q6: Providing the full prompt?**
>
> Yes and yes! In fact all of this is already included in the supplementary material that we submitted with the paper (but with possibly insufficient documentation, which we apologize for). You can see the full initial prompt in `get_sketch_prompt` of `eudoxus-main/src/eudoxus/llm/prompts.py` and the full hole filling prompt in `get_complete_prompt` of the same file. We are also happy to add the full conversations for every benchmark in the supplementary material along with a few in a new appendix (so that one does not need to execute the code to see a full conversation).

---

### Official Review · Reviewer_tGyy · 2024-07-14

**Soundness:** 4
**Presentation:** 4
**Contribution:** 3
**Rating:** 8
**Confidence:** 4

**Summary:**

The paper addresses the challenges of LLMs in generating code for very low-resource programming languages (VLPLs), which are not represented in their pre-training data. Traditional methods to enhance LLM efficacy in this domain include prompting, constrained decoding, and fine-tuning. The authors propose a novel approach called SPEAC, which leverages natural programming elicitation to align with LLMs' tendencies, including their common hallucinations. The technique involves translating a high-resource language (P) into a target VLPL (T) using a subset of P and iterative error repair through LLMs. The largest consistent sub-tree of a generated program in P is identified using a MAX-SMT solver, and errors are iteratively fixed. The approach is demonstrated on the UCLID5 verification language, showing significant improvement in syntactic correctness compared to traditional fine-tuning methods .

**Strengths:**

- The idea presented is novel, and the results are impressive, demonstrating significant improvements over traditional methods.
- The paper is well-written and engaging. The introduction clearly explains the contributions and the problem being addressed. The running example in Section 3 is helpful for understanding the proposed method.

**Weaknesses:**

I did not find any major weakness in the paper, however, the paper is focused on UCLID5 programs, which consist of a relatively narrow set of use cases, and it is not obvious how well this method will generalize to other VLPLs.

**Questions:**

In the paper, the authors propose using a subset of the high-resource language P, referred to as C, which can be transformed into the target low-resource language T. How is the existence of C determined in general?

**Limitations:**

Section 5.1 discusses the process of defining the subset C of P at a high level, but it lacks rigorous detail. I would appreciate seeing a more elaborate theory developed around this process in future work.

---

> ### Author Rebuttal · Authors · 2024-08-07
>
> We thank the reviewer for their thoughtful feedback. Please see below for our answers to the question posed.
>
> **tGyy-Q1: How is the existence of C determined?**
>
> Determining the existence of C is a design process. It is possible that some target languages are so esoteric that no subset of a high-resource language is even remotely compatible. However, showing that UCLID5, a language that is nothing like Python, has a good C in Python is evidence for the existence of a language C for many target languages.
>
> We described the general process for finding the child language C through grounded theory in Section 5.1 interleaved with the specific process for UCLID5 (mostly on lines 223 to 241) but we omitted some details due to space constraints. We will add detail to the description of the general process by drawing on general guidelines for using grounded theory in software engineering (e.g., Stol et al. 2016) and specific case studies (e.g., Barke et al. 2023). We will add detail to the specific process for UCLID5 by including concrete examples, possibly in the appendix.

---

> > ### Comment · Reviewer_tGyy · 2024-08-12
> > **Thanks for your response**
> >
> > Thank you for your response and explanation. Please include general guidelines and specific case studies in the next iterations of the paper. My question has been addressed, and I have no further inquiries.

---

### Decision · Program_Chairs · 2024-09-25

**Decision:**

Accept (poster)

**Comment:**

This work proposes a method for supporting LLM-based language generation in very low resource programming languages by invoking the LLM to generate code in a hallucinated library in a high-resource language instead, and then translating this program to the low-resource language using an SMT solver and iterative repair.

Reviewers agreed that this is both an important subject and that the proposed technique is quite novel and interesting. The results also highlight that this method can rather substantially improve the quality of the programs generated in a very low resource language.

At the same time, the reviewers noted that the evaluation is rather limited, focusing on just 33 (test) and 317 (fine-tuning) programs in one language (UCLID5). While the results on this set are very strong compared to baselines, this inherently provides limited evidence of the effectiveness and generalizability of the technique. Some reviewers also raised concerns around the limited choice of models and the absence of other prompting strategies as baselines.

As evidenced by the diverging scores, the reviewers weighed the limited evaluation against the technical novelty and positive results in different ways, with the majority opting to accept the work. Given this, and the importance of investigations in this under-studied domain and the significant novelty of this work, we have decided to recommend accepting this work. The authors are encouraged to incorporate the rich feedback from the reviews and to clearly note the limitations in the evaluation in the camera-ready version of the paper.